

# EvoNEST, a modular application for comprehensive species-based sampling and data management in comparative biology

Daniele Liprandi, Tom Illing and Jonas O. Wolff

Evolutionary Biomechanics Lab, Zoological Institute and Museum, University of Greifswald, Greifswald, Germany

Corresponding author
Daniele Liprandi,
daniele.liprandi@uni-greifswald.de

## ABSTRACT

Organism-based research in ecology and evolution often faces challenges in data management due to its commonly nested metadata structure (*e.g.*, phylogenetic, species, samples, subsamples and traits), fragmented data formats and limited accessibility and interoperability. Further, data migration between drives, platforms and formats can hinder data tractability, impede collaboration and risk critical data loss. Here we describe Evolutionary, ecological and biological Nexus of Experiments, Samples and Traits *(EvoNEST)*, a modular full-stack data management application designed to minimize data handling and conversions by integrating all aspects of biodiversity research, from specimen sourcing to publication, into a unified platform. Unlike traditional laboratory information systems that require extensive customisation for biological hierarchies, *EvoNEST*'s specimen-centric design natively supports taxonomic relationships and parent-child sample structures. We developed *EvoNEST Backbone*, a Docker-based core version of the software, which can be easily installed and customised for different research contexts. Its architecture, built on Node.js and MongoDB, ensures flexibility for diverse research needs while natively supporting Findability, Accessibility, Interoperability, Reusability (FAIR) data principles. Key features include intuitive graphical user interfaces, digital lab books, automated data import and export, trait analysis, integrated visualisation tools, and unique identifiers with QR codes for the seamless integration with physical objects (specimens, samples and subsamples). The system can be run locally or as a server-based application, facilitating data entry from mobile devices and computers within or outside the home institution. A logbook system assures that all changes to data and metadata are traceable, with live updates ensuring integrity when multiple users are accessing it. To demonstrate how *EvoNEST Backbone* can be customised to the researchers' needs, we present a specialised, network-enabled implementation dedicated to the comparative study of spider silk properties across spider species. We also discuss implementation choices and future developments. By providing a centralised, user-friendly system for organism-based data management, *EvoNEST* aims to accelerate the digitisation and dissemination of phenotypic data, and improving the traceability, documentation and reuse of the scientific output of laboratories. To ensures accessibility and to promote collaborative improvement, we release *EvoNEST* as an open-source software.

# INTRODUCTION

The field of ecology and comparative biology is increasingly data-intensive, with researchers collecting and processing information on organisms, their traits, and their environments (*Stucky et al., 2018*; *Perez-Riverol et al., 2019*; *Schoch et al., 2020*; *Pekár et al., 2021*; *Herberstein et al., 2022*; *Gloor et al., 2023*; *Bánki et al., 2023*; *Daru, 2024*). This rapid increase in data availability has substantially advanced our understanding of biodiversity, species interactions, and evolutionary processes. However, it has also presented substantial challenges in data management, particularly for research groups working with diverse species and the samples and traits derived from them (*Jones et al., 2006*; *Michener & Jones, 2012*).

Many laboratories and institutions in biodiversity research face common hurdles in data management: researchers often use a variety of tools and formats to record data, leading to inconsistencies and difficulties in data integration (*Michener, 2015*; *Poisot et al., 2019*); critical data is frequently scattered across different devices, file formats, and users, hampering collaboration and data retrieval (*Wilkinson et al., 2016*). As research projects grow in complexity, integrating interdisciplinary approaches that generate diverse data types, maintaining complete records of sample origins, treatments, and derived data becomes increasingly challenging (*Roche et al., 2015*). Without centralized and standardized data management systems, there is a greater risk of losing information due to human error or technical issues (*Hughes-Lartey et al., 2021*). Researchers often spend significant time on data entry, retrieval, and formatting, reducing time available for analysis and interpretation (*Boettiger et al., 2015*).

These challenges are particularly pressing in the current research landscape, where there are increasing demands for data digitization, complete documentation for regulatory compliance (*e.g.*, collection licenses, animal welfare legislation), and comprehensive data sharing for publication (*Whitlock, 2011*). This challenge has been discussed for more than fifteen years, highlighting how ecology could benefit from a systematic, better use of standardisation methods (*Kelling et al., 2009*; *Reichman, Jones & Schildhauer, 2011*; *Hampton et al., 2013*; *Powers & Hampton, 2019*; *König et al., 2019*). As an approach for the digital accessibility of data published in public repositories, in 2016, the Findability, Accessibility, Interoperability, Reusability (FAIR) principles were proposed (*Wilkinson et al., 2016*). These principles are now a fundamental part of open science practices and aim to provide guidelines to improve the findability, accessibility, interoperability, and reuse of digital assets (*Wilkinson et al., 2016*). While literature exist on how to apply FAIR data principles to large cloud databases (*Mons et al., 2017*; *Jacobsen et al., 2020*; *van Reisen et al., 2020*; *The Galaxy Community et al., 2022*), there is a lack of discussion about how local data could be prepared in each ecology and comparative biology laboratory according to those principles (*Borer et al., 2009*; *White et al., 2013*), while more general works highlight how open data often does not have the necessary data quality (*Nikiforova, 2020*). Often a variety of files, especially calculation sheets (such as Excel spreadsheets) structured in individual, non-standardized ways, are used by different lab members. These sheets are

then adapted and restructured several times depending on the current need, *i.e.*, an analysis, a data exportation, or an upload to an online database.

To address these challenges, we have developed Evolutionary, ecological and biological Nexus of Experiments, Samples and Traits *(EvoNEST)*, a modular full-stack application designed to streamline the collection, management, exploratory analysis, and preparation for publication of data in ecological and evolutionary research. EvoNEST provides a centralized system for storing, processing, and analysing organism (specimen, sample, and subsample) data, measurements, and raw data files. *EvoNEST* aims to improve researchers' efficiency, while ensuring that the data natively respect the FAIR principles.

Although initially developed for a comparative study on spiders and spider silk properties, EvoNEST has been designed with flexibility in mind, allowing for adaptation to a wide range of organism-based research contexts. The platform can accommodate various types of organisms, from plants to animals, and can be customised to handle different types of samples, traits, and experimental data. To facilitate adoption across different research contexts, we provide a backbone version of *EvoNEST* that can be easily installed and customised while maintaining all core functionalities.

Compared to already existing platforms sharing similar missions (*Gries, Gilbert & Franz, 2014*; *Popović et al., 2020*; *Bogan et al., 2023*), *EvoNEST* provides a local solution which can be installed within the digital boundaries of the laboratory. *EvoNEST* focuses on providing the tools to optimise not only the collection, but also the management and analysis of the organisms. These advantages allow new and existing laboratories to standardise their local data management. The data architecture does not have a rigid schema, allowing facilities to customise their own platform and data structure, share their data, while keeping ownership of their data. A similar approach has been successfully implemented before by *OpenBioMaps*, a data management platform dedicated to spatial biodiversity data (*Bán et al., 2022*).

In this article, we present the architecture and key features of *EvoNEST*, demonstrating how it addresses common challenges faced by comparative biologists collecting samples and data across many species. We outline how storing and managing data through *EvoNEST* allows for a more transparent data handling, and how *EvoNEST* implements FAIR data principles. Finally, we compare *EvoNEST* with existing solutions, and we present, as a case study, an implementation based on a comparative study on spiders.

## METHODS AND SOFTWARE OVERVIEW

*EvoNEST* is a database creator, manager and visualiser full-stack application based on Node.js (https://nodejs.org/). Like any Node.js application, it can be run locally on a machine, or deployed on a server. In both cases, it is accessible and navigable *via* a browser. *EvoNEST*'s frontend is built using the Next.js framework. The backend is written using Next.js (https://nextjs.org/) and MongoDB (https://www.mongodb.com/). Data visualisation is granted by a variety of data visualisation tools, including customisable plots and tables, based on both self-written components and external libraries, cited in our

**Figure 1** **Schematic illustration of the back-end and front-end of *EvoNEST*.** The back-end consists of a MongoDB database with three main collections (Samples, Experiments, and Traits), each with associated logbooks for change tracking. The database interfaces with external files and provides data access through a Read/Write RESTful API and local export capabilities. The front-end presents data through interactive tables and visualization tools, with direct connections (pink arrows) to the corresponding database collections. Different NESTs can be selected easily from the selector appearing in the navigation bar. Analysis tools in the front-end provide real-time data exploration and visualization capabilities.

repository. A schematic overview about the software design is found in Fig. 1. The entire architecture is containerized using Docker, allowing rapid installation and customisation (*Merkel, 2014*).

## System architecture

The software follows a three-tier architecture: a presentation tier (frontend), an application tier (backend), and a data tier (database). The frontend is implemented with Next.js, providing a responsive user interface accessible from different devices. The backend, also implemented with Next.js, handles data processing, application programming interface (API) routing, and authentication with NextAuth. The database layer utilises MongoDB, with files stored in a dedicated storage volume.

## Database structure

To avoid data fragmentation, the database managed by *EvoNEST* includes four collections: *samples*, *experiments*, *traits*, plus the auxiliary *files* collection. We call a database inside of *EvoNEST*, *i.e.*, all the collection together with the uploaded files, a NEST. One instance of *EvoNEST* supports multiple NESTs.

The *samples* collection stores information about biological specimens, including taxonomic classification, collection metadata, physical storage location, and hierarchical relationships to other samples. Each sample document contains fields for species identification, collection date and location, responsible researcher, and sample type (*e.g.*, animal, plant, silk).

The *experiments* collection stores data about procedures performed on samples, linking raw data files with metadata about experimental conditions. This collection includes fields for experiment type, date, parameters, and results, with references to the sample identifiers used in the experiment.

The *traits* collection stores processed measurements derived from samples or experiments, including both direct measurements and calculated values. Each trait document includes fields for the measurement value, unit, measurement method, and references to the sample identifiers.

A fourth auxiliary collection, *files*, stores metadata about uploaded files, including their location in the file system, file type, and relations to samples, experiments, or traits. The actual files are stored in a dedicated file storage volume, with their paths recorded in the database.

Each of these collections can be managed by the administrators, using JavaScript scripts or a mongoexpress interface (https://github.com/mongo-express/mongo-express). Each collection transaction is recorded at the entry level, by storing the date, user, and action in a custom field "logbook", which encompasses the history of the data entry. Each data entry has a unique identifier, which is used as a universal key for the entry itself. This allows *EvoNEST* to generate QR (quick-response) codes to link physical specimen containers with their corresponding digital records. These QR codes encode the unique identifiers of the samples inside the database. Several functions depending on the sole specimen identifier can be implemented into *EvoNEST*, including opening the relevant specimen page in *EvoNEST*'s web interface, and tracking general maintenance tasks, such as animal feeding, plant watering. Using the camera of the device (mobile phone or computer), these functions are able to scan the QR codes to immediately retrieve the correct identifier and streamline the maintenance work.

MongoDB stores its data natively in a BSON (Binary JSON) format, MongoDB custom format. We deliberately opted not to impose strict schemas or key constraints on our database, avoiding tools such as Mongoose. This allows each user group and admin to have the maximum flexibility in how to manage the NEST. To enable integration with external systems, *EvoNEST* implements a RESTful API that provides programmatic access to the database collections. The API follows standard HTTP methods (GET, POST, DELETE) for data operations and returns responses in JSON format. All API endpoints are documented using the Swagger format (https://swagger.io/). The full API documentation is available, thanks to a Redoc parser, inside the application, at the local URL/api-docs.

## Features

*EvoNEST* core features include: sample and subsample management, customisable sample and trait types for each database *via* user interface (UI), collection metadata tracking (sampling dates and locations, collectors, and specimen storage locations), hierarchical linking between samples, custom data tables for each collection, custom multiple databases with different access rights for each user, individual pages for each entry with editable fields, automatic and customisable trait statistical analysis, and geolocation mapping of collection sites. These features form the foundation of *EvoNEST*'s data management

**Table 1 Summary of features available in *EvoNEST* at the time of writing.**

| Feature | Description |
|---|---|
| Data model | Flexible and extensible structure accommodating diverse research needs. Hierarchical organization of data (organisms, samples, derived data) with custom field options. Support for the allocation on the database of direct and derived measurements, together with their associated metadata and files. |
| NESTs | Supports multiple databases (NESTs) inside the same application, with customisable sample and traits categories for each NEST. |
| User interface | Customizable dashboard displaying recent activities and key metrics. Detailed specimen views provide comprehensive information about individual organisms and associated data. |
| Search functionality | Advanced search and filter capabilities through the use of custom columns component, which can be personalised by the NEST administrator. |
| QR codes | Unique QR codes generation. Support for the lookup and adding of specimen (meta)data, as well as logging of maintenance function through the use of QR codes, *i.e.*, feeding of animals, tank cleaning. |
| Import options | Import samples from externally generated files. Support hierarchical upload, using auto-recognition of specimen names to upload respective specimens and specimen parents from a single table. Native support for data logger files from STAR-ODDI and LoggerMate sensors. |
| Export options | Support for CSV, Excel, and JSON formats, with API access for programmatic data retrieval. |
| Data visualization and analysis | Interactive charts and basic statistical functions for data exploration. Trait analysis page allows the user to computes statistical measurements of grouped data according to customisable filters. |
| Metadata recording | Comprehensive metadata capture ensuring data provenance, including from images, text documents and manual collected data. User edits and actions for each entry are stored in logbooks, navigable by the user by date. |
| External database integration | API-based integration with databases such as the Global Biodiversity Information Facility (GBIF) for automated retrieval and validation of taxonomic information. |
| Geolocation services | Integration with OpenStreetMap for accurate recording and visualization of collection locations. |
| Security measures | User authentication through NextAuth, data activity logging. |

capabilities, designed to support various research workflows from field collection to data analysis. *EvoNEST* also natively supports data and metadata extraction from logger files of a variety of sensors from LoggerMate (https://www.loggermate.com/) and STAR-ODDI (https://www.star-oddi.com/), including GPS trackers, accelerometers, temperature sensors, *etc*. To speed up data entry and improve the user experience, we implemented several courtesy functions, such as the one-click retrieval of GPS coordinates from a mobile device, the lab location and maintenance logbook entries (*e.g.*, feeding, observation of moulting and death). Further, it is possible to connect to the camera of the mobile device and directly add photos to sample entries with one click, or to quickly access sample entries by scanning the QR code on the physical specimen. Specimen labels containing the QR code and customised specimen information can be generated and prepared for printing inside *EvoNEST*. A collection of currently implemented features is presented in Table 1.

## Fair data principles

The FAIR principles provide general guidelines for making research data Findable, Accessible, Interoperable, and Reusable (*Wilkinson et al., 2016*). These principles are increasingly important for biodiversity research, where data integration across studies can reveal large-scale patterns and processes (*Gurevitch, Curtis & Jones, 2001*; *König et al., 2019*; *Gallagher et al., 2020*; *Herberstein et al., 2022*). *EvoNEST* provides the infrastructure to easily implement these principles by facilitating data harmonisation, tractability and

publication as outlined below. We note that, while *EvoNEST* provides a reliable way to comply to these principles inside a single project and facilitates the preparation of datasets for the FAIR compliant publication, it is important to check again the compliance to FAIR principles when the data is deposited in an open online repository for the long-term access (*e.g.*, compliance with the specific repository standards).

### F1. (Meta)data are assigned a globally unique and persistent identifier

Each user, sample, experiment and trait of a NEST has a global unique identifier, generated inherently by MongoDB and stored in the _id field of each entry. To comply this principle, we used the intrinsic MongoDB identifier generator to generate a 12 byte string for each entry of the database.

### F2. Data are described with rich metadata (defined by R1 below)

We implemented the possibility to enter different types of metadata for each data entry. *E.g.*, for biological samples, we enforce the insertion of the collection date and location, name of the collector, species name, and sample type. The framework permits the addition of other possible metadata. MongoDB also allows for the insertion of nested metadata, thanks to its use of BSON data format.

### F3. Metadata clearly and explicitly include the identifier of the data they describe

All metadata are inserted in the same row of the database of the unique identifier. Metadata in different tables contain, in the same row, the entry identifier referring to the original data row.

### F4. (Meta)data are registered or indexed in a searchable resource

All (meta)data are stored in machine-readable format inside the MongoDB database, which is indexed and searchable using MongoDB native indexing system, stored in the _id field of each entry. All data can be accessed directly by using the open API defined for each collection. The API use the RESTful standard, and data can be accessed using the GET operation.

### A1. (Meta)data are retrievable by their identifier using a standardised communications protocol

All (meta)data are retrievable using JavaScript and/or the API provided with *EvoNEST*. Files associated with the data entries are stored on the machine where *EvoNEST* is running. The file is renamed using a predefined format chosen by the admin. The filepath attribute contains the location of the file inside the project, and hence files can be easily accessed through the database entry instead of searching through a folder-subfolder structure.

### A2. Metadata are accessible, even when the data are no longer available

Metadata of files are always stored directly in the database when a file is inserted, and thus do not need the existence of the file themselves. This applies both for data stored directly in the database and for large data stored externally.

*I1&I2. (Meta)data use a formal, accessible, shared, and broadly applicable language for knowledge representation/(Meta)data use vocabularies that follow FAIR principles*

(Meta)data are stored using only open-access non-proprietary format. All literal and numerical data are stored directly inside the MongoDB database. The definition of different types of samples, traits, and other features, are stored statically, and are editable by the administrator by following the instructions including in the repository. All users can see the vocabularies from the graphical user interface. In the current version of *EvoNEST* it is the responsibility of the NEST administrator to identify and apply common standards and definitions of traits, methods and taxonomic nomenclatures to their data. We plan to add automatic conversion to common vocabularies, *e.g.*, ontologies and thesauri (*Jonquet et al., 2011*; *Buttigieg et al., 2016*), in the future.

*I3. (Meta)data include qualified references to other (meta)data*

As per principle F1 and F3, (meta)data always refer to other (meta)data by using the unique identifiers.

*R1. (Meta)data are richly described with a plurality of accurate and relevant attributes*

*EvoNEST* aims to collect all the (meta)data regarding a single sample, measured trait, experiment or file identified as useful by the user.

## Code distribution and installation

*EvoNEST* is distributed as a backbone version, called *EvoNEST Backbone*, providing all the core functionalities described in Table 1. *EvoNEST Backbone* is distributed both as a Docker image, and as an open repository. The open repository includes Docker containerization files, API documentation, user guides, and developer resources. A permanent snapshot of the codebase at the time of publication has been deposited on Zenodo (https://doi.org/10.5281/zenodo.15056682), while current Docker images are distributed through Docker Hub. The installation of *EvoNEST Backbone* is streamlined through Docker Compose, requiring minimal to none technical expertise about JavaScript environments. Using the source code and the provided Docker configuration files, it is also possible to customise the platform according to specific needs, such as modifying the user interface, changing the user authentication system, or implementing custom data processing workflows. *EvoNEST Backbone* can be deployed either locally (accessed through localhost) or as a network-accessible application through Docker containerization.

For local deployment, *EvoNEST* runs on a single machine, suitable for individual researchers or small teams sharing a work desktop. Network deployment enables broader access and collaboration through a custom domain, making the application available across multiple devices and locations. Both configurations use the same underlying architecture, differing only in their access scope and authentication requirements. While it comes with default authentication settings, users aiming to install *EvoNEST Backbone* on a

web-accessible server are encouraged to implement their preferred authentication provider for improved security.

The complete *EvoNEST* source code, including the backbone version, customisation examples, and documentation, is available in the project's GitHub repository at https://github.com/daniele-liprandi/EvoNEST-backbone. The repository also includes the user and developer guides, both available at https://daniele-liprandi.github.io/EvoNEST-backbone/. We note that some expertise in network administration and coding is required to enable network access for the NESTs, *e.g.*, using nginx, and to customise *EvoNEST* beyond the acceptance of custom sample, subsample and trait types.

## CASE STUDY: COMPARATIVE STUDY OF SPIDER TRAITS

To demonstrate *EvoNEST*'s capabilities, we show a fully implemented NEST focusing on comparative spider research. This implementation showcases both the backbone version's core functionalities and customised features developed for spider and silk research. A simplified entity-relationship diagram of this implementation is shown in Fig. 2.

Building upon the foundations of the backbone version, we developed specialized features for spider and spider silk research. We implemented automatic taxonomic data retrieval and validation through integration with the World Spider Catalog (WSC) online database. This not only allows for the automatic addition of higher taxonomic entities (*e.g.*, family assignments), and the automatic correction of misspelled species names, but also the automatic update of taxonomic changes to sample species. The subsample management system was customised to handle silk samples, with specialised categorisation by silk sample type (*e.g.*, dragline, capture thread). Usually, multiple silk samples can be derived from a single animal, and, further, it is possible that silk samples are further divides into silk subsamples for different analyses, all of which is accounted for by *EvoNEST*'s nested database structure.

For the import and processing of experimental data, we extended the backbone's experiment collection capabilities to process output files from a KLA T-150 universal testing machine that was used to measure the mechanical properties of spider silk samples. Custom modules automatically extract and calculate key mechanical properties such as peak strength, extensibility, and toughness, and store these as traits. Traits acquired through direct measurements (such as the body mass of spiders) can also be added manually to the trait list. In addition, diverse files generated through observations and experiments, such as documents and images, can be stored and connected with sample and trait entries. The dashboard and some visualisation tools were customised to highlight important traits and trait correlations measured in the lab and, to enable project monitoring and exploratory data analyses.

This implementation is available as a demo at https://evonest.zoologie.uni-greifswald.de/. In a version of this application, currently used in our lab (Fig. 3), twelve active users contribute to the main laboratory NEST, managing at the time of writing over 4,100 samples, 700 experiments, and 7,800 traits, demonstrating the system's scalability for specialized research contexts.

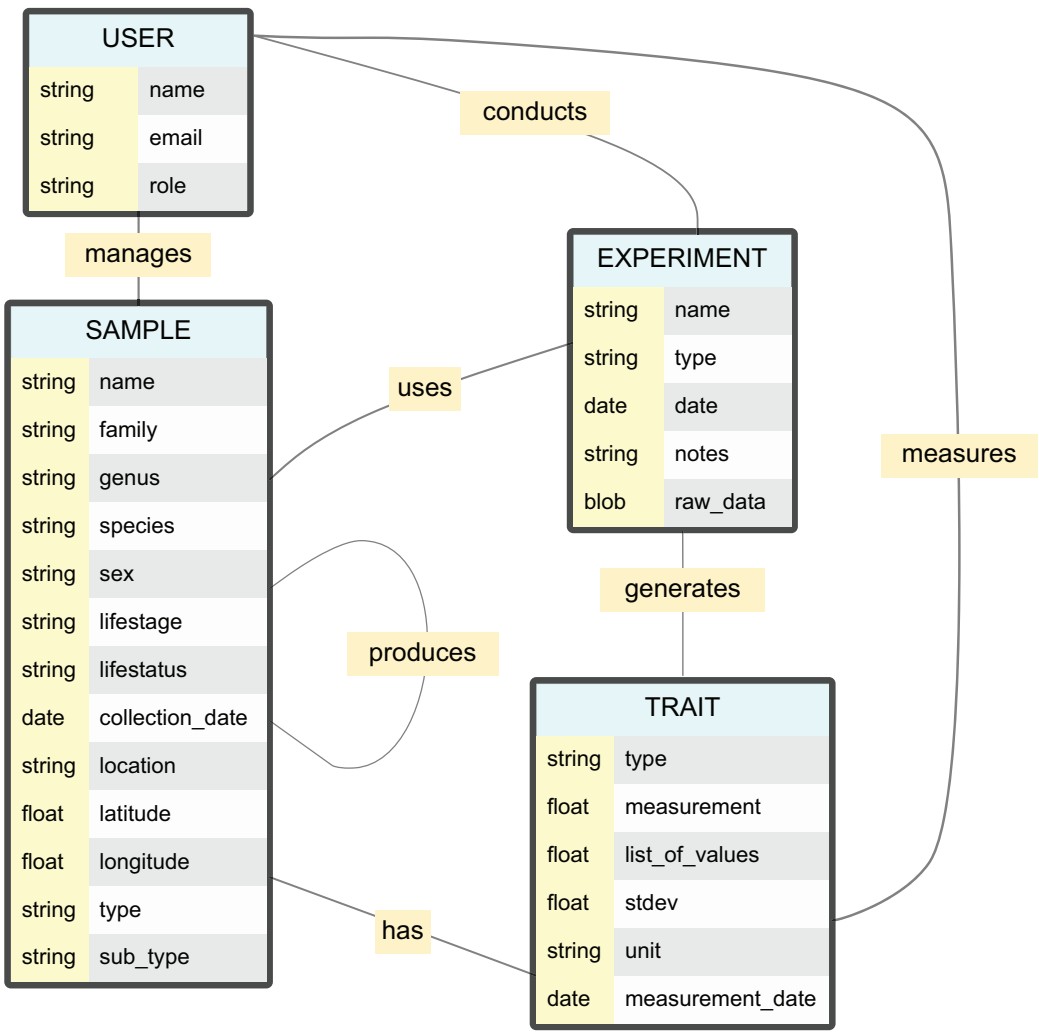

**Figure 2 Entity relationship diagram for the NEST used in the case study.** Each box represents a collection in the database, with main field names and their data types listed. The USER collection manages researcher information and permissions. The SAMPLE collection stores specimen data including taxonomic classification, collection metadata, and physical location. The EXPERIMENT collection handles raw data and experimental parameters, while the TRAIT collection stores processed measurements and derived data. The FILE collection, not included here, is connected to all other collections and stores files metadata.

## COMPARISON WITH EXISTING SOLUTIONS

Several systems have been developed to address challenges in biodiversity and laboratory data management. Known solutions are, for example: Electronic Lab Notebooks (ELN) (*Kanza et al., 2017*), LIMS (Laboratory Information Management Systems) applications for biodiversity research, self-deployable solutions like *OpenBioMaps* (*Bán et al., 2022*), and museum-focused platforms such as *Symbiota* (*Gries, Gilbert & Franz, 2014*). However, *EvoNEST* addresses a specific gap in this ecosystem that existing systems do not effectively fill. A comparison between *EvoNEST* and existing solutions is available in Table 2.

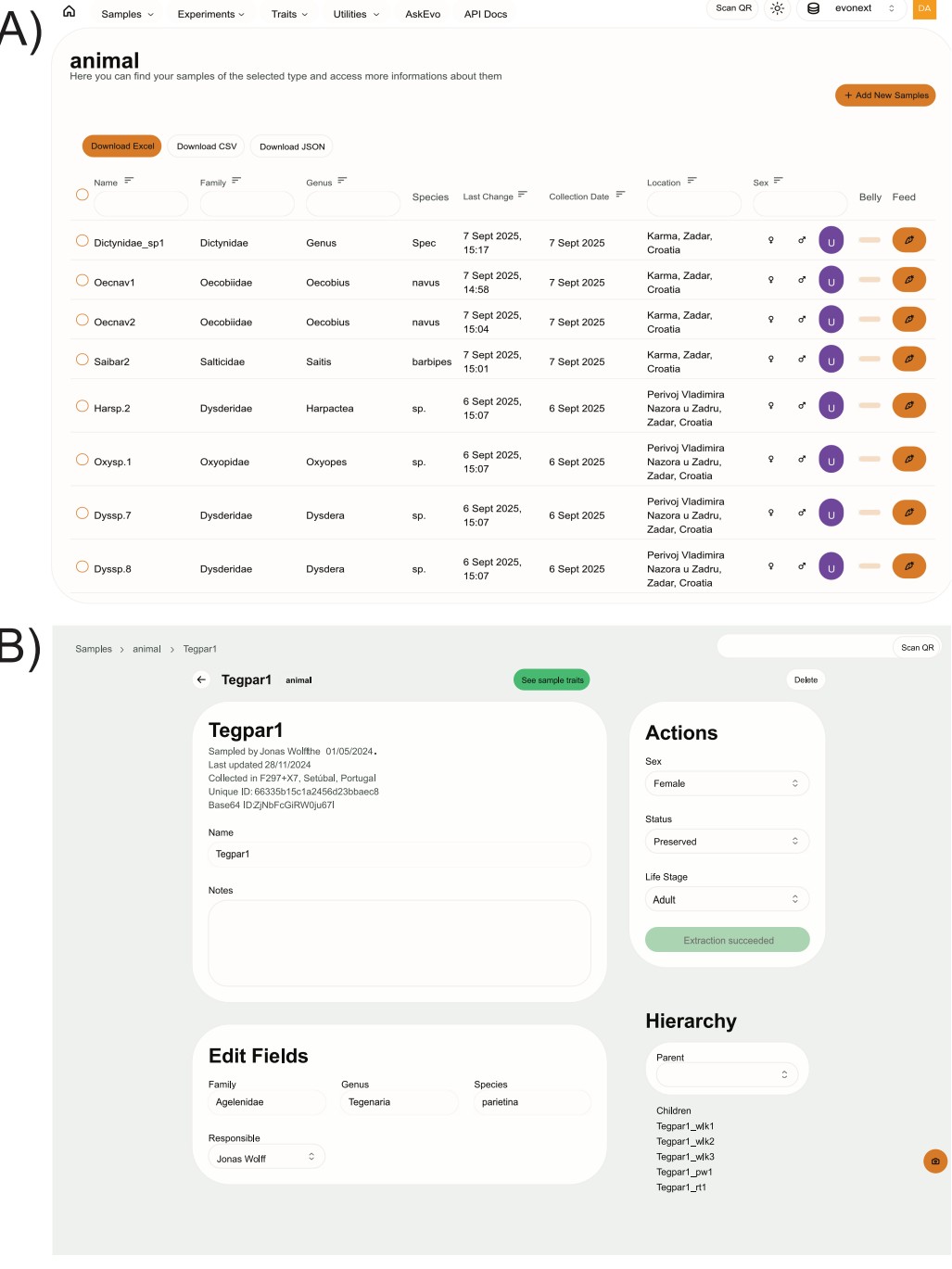
**Figure 3 Example screenshots of webpages for the case study discussed in this section.** (A) Data table for the animals. The columns *Name*, *Family*, *Genus*, *Location* and *Sex* are searchable and filterable. The whole or filtered datasets can be downloaded in common spreadsheet formats. Interactive buttons can be used by the users to quickly edit data and add maintenance records to animal entries. (B) Single sample webpage. Different information is available through the use of cards, which can be customised by the NEST administrator. The shown data for both examples are retrieved using the open API.

**Table 2 Comparison between *EvoNEST*'s features and currently existing solutions at the time of writing.**

| Feature category | *EvoNEST* | Traditional LIMS | Electronic lab notebooks | *OpenBioMaps* | *Symbiota* |
|---|---|---|---|---|---|
| **Primary focus** | Laboratory organism research | Clinical/Manufacturing workflows | Research documentation | Spatial biodiversity monitoring | Museum collection management |
| **Technical architecture** | | | | | |
| Database backend | MongoDB (document-based) | Relational (SQL Server, Oracle) | Varies by vendor | PostgreSQL/ PostGIS | MySQL |
| Deployment | Docker/self-hosted | Enterprise installation | Cloud/on-premise options | Docker/self-hosted | Web hosting |
| **Taxonomic integration** | | | | | |
| Taxonomic integration | Native (WSC, GBIF APIs) | Custom development required | Manual entry | GBIF integration | Taxonomic authorities |
| Hierarchical taxonomy | Built-in organism relationships | Requires schema customization | Free-text documentation | Occurrence-based | Strong taxonomic tools |
| **Laboratory workflow** | | | | | |
| Equipment integration | Customisable file processing. Native support foro LoggerMate and STAR-ODDI sensors. | Vendor-specific modules | Limited connectivity | Not applicable | Basic specimen tracking |
| Specimen maintenance | Strong and customisable | Basic sample tracking | Manual logging | Not applicable | Curation workflows |
| Trait data, sample data and meta-data management | Native processing & analysis | Configurable workflows | Documentation-focused | Limited support | Specimen metadata |
| **Collaboration & access** | | | | | |
| Multi-user support | Built-in real-time collaboration | Enterprise user management | Platform-dependent sharing | Multi-user editing | Web-based collaboration |
| Mobile interface | Responsive design | Limited mobile support | Variable by vendor | Responsive design | Basic mobile access |
| **Cost Structure** | | | | | |
| Initial cost | Free (open-source) | $3,000–$250,000+ annually | $0–$50,000+ annually | Free (open-source) | Free (open-source) |
| Implementation | Community/self-support | Enterprise IT required | Vendor-dependent | Community/self-support | Community/self-support |
| **Regulatory compliance** | | | | | |
| Data validation | Configurable rules | GxP/FDA compliance focus | Basic audit trails | Research data standards | Collection standards |
| Audit trails | Automatic change logging | Enterprise audit systems | Variable by platform | Version control | Basic change tracking |

Traditional LIMS platforms focus primarily on sample tracking and workflow management in clinical or manufacturing environments, with licensing costs ranging from $3,000 to $250,000+ annually and implementation complexity requiring dedicated IT resources. While powerful for enterprise environments, these systems require extensive customisation to handle organism-based research workflows and taxonomic hierarchies that are fundamental to biological collections. Electronic Laboratory Notebooks (ELNs) provide flexible documentation capabilities and offer many benefits over physical lab

notebooks, including networked digital environments, collaboration support, and contribution to Good Research Practice through tracking and documentation of research processes (*Vandendorpe et al., 2024*). However, as ELNs are primarily designed as digital replacements for traditional laboratory notebooks rather than extensive data management systems for complex specimen relationships and taxonomic hierarchies, they face inherent limitations when applied to systematic organism-based research. Researchers often maintain separate systems for specimen tracking and experimental data analysis, creating parallel data streams with attendant risks of inconsistency and duplication (*Vandendorpe et al., 2024*). Unlike these platforms that focus primarily on sample tracking and workflow management, *EvoNEST* was designed from the ground up for organism-based research with inherent support for taxonomic hierarchies, parent-child relationships between specimens, and the complex web of derived samples and measurements.

More biodiversity-specific solutions are rich in features and require less training, but lack flexibility and are often more suited for collections rather than active experimental research environments: *OpenBioMaps* and *Symbiota* serve, respectively, the spatial biodiversity monitoring community, and natural history museums and herbaria. While each software excels in specimen cataloguing, taxonomic databases, and collaborative digitization, they lack integrated living culture and specimen maintenance tracking, document data processing, and the trait analysis capabilities required for organism-based research.

## DESIGN CONSIDERATION AND DEVELOPMENT ROADMAP

The development of *EvoNEST* involved several technical decisions that shaped its current implementation. The decision with the greatest impact on the application development regarded the data structure. We chose MongoDB over relational databases to accommodate heterogeneous data types with variable attributes across different organisms and research contexts. This document-oriented approach allows researchers to add new fields or data types without restructuring the database, supporting the exploratory nature of ecological and evolutionary research. Although MongoDB's document-oriented approach introduces some computational overhead compared to optimised relational queries, we determined this cost to be minimal relative to the flexibility gains. This flexibility has proven valuable for diverse research applications, as it allows developers to quickly create new front-end components dedicated to the management of their specific organism. For each modular component of *EvoNEST*, we provide template files that can be used as reference. This is described in our developer guide, available at https://daniele-liprandi.github.io/EvoNEST-backbone/developer-guide.

We plan to further enhance *EvoNEST* through several upcoming developments, including:

- Standardised data exchange formats and direct integration with major biodiversity repositories. This will include semi-automated data submission to platforms like Global Biodiversity Information Facility (GBIF), through the use of custom data transformers and, when available, repository APIs.

- Specialised front-end modules for different research domains (*e.g.*, plant phenotyping, microbial cultures, longitudinal ecological studies).
- Implementation of locally-installed large language models (LLMs) to provide contextual assistance and data exploration capabilities. This will allow non-expert users such as students and early-career researchers to query complex datasets using natural language, generate preliminary analyses, and receive guidance on data collection protocols without requiring advanced database or programming skills.

To continue *EvoNEST*'s development through community involvement, we have adopted an open-source strategy. Collaborating on the development of EvoNEST is possible through the *EvoNEST Backbone*'s repository, at the URL https://github.com/daniele-liprandi/EvoNEST-backbone. The user guide, the developer guide, and a static version of the API Docs are all available at https://daniele-liprandi.github.io/EvoNEST-backbone/. We are establishing a community-driven expansion model where researchers can contribute specialised modules or modifications tailored to specific research contexts. This approach has proven successful in other scientific software ecosystems, such as R/Bioconductor (*Huber et al., 2015*) and Galaxy (*The Galaxy Community et al., 2022*).

## CONCLUSIONS

We have shown here that *EvoNEST* has broad applicability across various domains of ecological and evolutionary research. It is particularly suited for comparative biology studies that collect data across organisms and species, as well as long-term ecological monitoring and integrative biodiversity research. The platform's versatility allows researchers to use it for a wide range of organisms, including microorganisms, plants and animals, in both field and laboratory settings. While basic knowledge of server management is beneficial for network deployment, the backbone version's Docker-based installation process makes *EvoNEST* accessible to users with minimal technical expertise. Once installed, the system can be operated and customised by users without a programming background.

One of the main features of *EvoNEST* is its schema-less database design. We chose to follow this strategy to satisfy two needs of organism-based research: the existence of a multitude of data structures, and the growing necessity of organising and preserving the data. Both these needs exist in active research projects and when preparing data for publication, as the scientific community and the funding agencies require more and more transparency over the years. While this design choice sacrifices some computational efficiency when compared to more structured databases, these costs are offset by the ease of use that is gained. This allows *EvoNEST* to be easily implemented in every laboratory, without requiring the large amount of ad-hoc customisation needed by other existing alternatives, as ELNs and LIMS.

The goal of our application is to democratise access to biodiversity data, preserve data privacy and ownership, and increase the efficiency of laboratories dealing with different types of organismal data. To achieve these objectives, we invite interested scientific communities to engage with *EvoNEST*. Scientists and developers can contribute to the

development of the app by providing feedback and feature requests, or by helping directly in the development of the app or its plugins and modules. All these actions can be taken through our GitHub repository. A collaborative approach to the development of *EvoNEST* will ensure a continuous evolution of its features, allowing it to meet the diverse and changing needs of the research community.

As data management challenges in biodiversity, ecology and evolution continue to grow, tools like *EvoNEST* give researchers the means to effectively organise, analyse, and share their data. Furthermore, having a unified platform for data collection, data storage and organism maintenance helps laboratories establish collaborative experiments, without sacrificing the ability to quickly run their analysis. By aiding the digitisation of phenotypic data, a digitisation that is still lagging behind the one of genetic data, *EvoNEST* may contribute to the advancement of our understanding of biodiversity and evolutionary processes.

## ACKNOWLEDGEMENTS

Daniele Liprandi thanks all the people of the Thursday morning meetings at the Computer Centre of Universität Greifswald for their valuable technical advice, tips and recommendations. Daniele Liprandi also thanks STAR-ODDI (https://www.star-oddi.com/) and LoggerMate (https://www.loggermate.com/) for providing copies of the output files from their animal sensors and custom data loggers for free, making the development of the parsers possible. Daniele Liprandi and Tom Illing have used Anthropic Claude to review grammatical accuracy and logical coherence of the manuscript.

### Funding
This study was funded by a European Research Council Starting Grant of the European Union (101040724—SuPerSilk). The funders had no role in study design, data collection and analysis, decision to publish, or preparation of the manuscript.

### Grant Disclosures
The following grant information was disclosed by the authors:
European Research Council Starting Grant of the European Union: 101040724.

### Competing Interests
The authors declare that they have no competing interests.

### Author Contributions
- Daniele Liprandi conceived and designed the experiments, performed the experiments, analyzed the data, performed the computation work, prepared figures and/or tables, authored or reviewed drafts of the article, and approved the final draft.
- Tom Illing performed the computation work, authored or reviewed drafts of the article, and approved the final draft.

- Jonas O. Wolff conceived and designed the experiments, authored or reviewed drafts of the article, and approved the final draft.

## Data Availability

The entire codebase and an importable docker-image are available at GitHub and Zenodo:

- https://github.com/daniele-liprandi/EvoNEST-backbone.
- Liprandi, D. (2025). EvoNEST backbone (v1.1.0). Zenodo. https://doi.org/10.5281/zenodo.15880272.

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
