# Peer review of "EvoNEST, a modular application for comprehensive species-based sampling and data management in comparative biology"

_PeerJ Computer Science, doi:10.7717/peerj-cs.3186_

## Round 0.1 · original submission · Minor Revisions

Please review your manuscript by following all requests and suggestions of the reviewers.

·

Basic reporting

The introduction section lacks a review of recent works on the topic. By the way, authors compare EvoNEST with other software in section 3.1.
L147: Is there API documentation available?

Experimental design

L139: The usage of QR codes seems unclear despite the description.
L169: Only Docker container is available; the source code would be helpful to keep the software up to date. Docker Hub is more suitable for software releases than Figshare, Zenodo, etc repositories.
L332: While the open-source approach has been chosen, the GitHub/GitLab repository is strongly recommended.

Validity of the findings

The user manual is recommended on the EvoNEST website https://evonest.zoologie.uni-greifswald.de/

Additional comments

Minor text issues:
L136: missing “)”
L226: missing reference

Reviewer 2 ·

Basic reporting

-

Experimental design

-

Validity of the findings

-

Additional comments

1. It is stated that the Evolutionary, Ecological and Biological Nexus of Experiments, Samples and Traits (EvoNEST), as described in the study, is a specimen-centric, open-source, and modular data management system developed to address challenges related to hierarchical data structures, fragmented formats, and limited accessibility commonly encountered in organism-based ecological and evolutionary research; and that the system aims to enhance data traceability, interoperability, and reusability by integrating the entire research process—from specimen collection to publication—within a unified platform.

2. In the introduction, what the developed Ecological and Biological Nexus of Experiments, Samples and Traits is, and the importance of the subject are mentioned at a basic level. Although the explanations in this section are generally appropriate in relation to the study, at the end of this section, it is necessary to specify in more detail the fundamental differences of the modular application developed within the scope of the study from the literature and the innovations it brings to the literature, and the originality sections.

3. The types of features specified in Table 1 and the explanations corresponding to them regarding the features in Ecological and Biological Nexus of Experiments, Samples, and Traits are both sufficient and explanatory.

4. The Schematic illustration in Figure 1 and the Diagram consisting of the trait, sample, user, and experiment sections specified in Figure are at a suitable level for the study, and the quality of the study is clearly demonstrated.

5. It is suggested to add a table so that this study can stand out more in relation to comparison with existing solutions. In addition, adding a detailed table regarding the literature will make the study stand out more.

As a result, although the modular application literature within the scope of this study has the potential to make a very important contribution, attention should be paid to the sections specified above.

---

## Round 0.2 · accepted · Accept

The authors addressed all concerns and requests adequately.

Reviewer 2 ·

Basic reporting

All comments are in the last section.

Experimental design

All comments are in the last section.

Validity of the findings

All comments are in the last section.

Additional comments

I find the authors’ responses to my comments and the corresponding revisions to be sufficient. I have no further suggestions for correction. Best regards.